**Data Availability Statement:** All relevant data are within the manuscript.

**Funding:** The funders had no role in study design, data collection and analysis, decision to publish, or

# Funcionality assessed by the core set of the international classification of functionality and health for rheumatoid arthritis: A cohort study

**Juliana Portes** [1,2,3]*, **Rafaela C. E. Santo**[1,2,3], **Ricardo M. Xavier**[1,2,3], **Claiton V. Brenol**[1,2,3]

1 Hospital de Clínicas de Porto Alegre, Laboratório de Doenças Autoimunes, Porto Alegre, Brazil,
2 Universidade Federal do Rio Grande do Sul, Faculdade de Medicina, Porto Alegre, Brazil, 3 Hospital de Clínicas de Porto Alegre, Serviço de Reumatologia, Porto Alegre, Brazil

* julianaportes24@gmail.com

## Abstract

### Objective

The aim of this study was to evaluate the function of a cohort of patients with rheumatoid arthritis (RA) from the core set of the International Classification of Functioning and Health (ICF) for RA over 12 months.

### Methods

We used prospective longitudinal data to conduct a cohort study among a well-characterized group of RA patients. Ninety RA patients aged between 40 and 70 years were included in the study. Patients were evaluated at baseline and after 12 months. Age, disease duration, current smoking, erosions, disease activity, functional test, disability and physical activity were evaluated. Then, the ICF core set classification for RA was applied.

### Results

81 patients completed the assessments, the majority of patients were female (88.9%) and the mean age was 56.5 ± 7.3 years. At baseline, the median disease activity was 3.0. There was a statistically significant (p < 0.02) improvement in "Exercise tolerance functions" over 12 months and also a statistically significant (p < 0.001) decrease in "Muscle strength functions" over 12 months. The activity and participation domain showed a weak correlation with the clinical data of the DAS28-PCR (p<0.02).

### Conclusion

We conclude that relevant aspects of the ICF Core Set for RA were able to adequately express the physical and functional factors of the RA cohort. This tool provides a common language for the interdisciplinary team, which can enhance the use of timely interventions to prevent physical disability in clinical practice.

preparation of the manuscript. This study was funded by the Research Incentive Foundation (FIPE) of the Hospital de Clínicas de Porto Alegre and the Coordination for the Improvement of Higher Education Personnel (CAPES). FIPE was used to cover general research expenses and a CAPES grant was given to study author Juliana Portes. Funders did not participate in any stage of the study. https://www.gov.br/capes/pt-br https://www.fipe.org.br/pt-br/ensino/extensao-curta-duracao/.

**Competing interests:** The authors have declared that no competing interests exist.

## Introduction

Rheumatoid arthritis (RA) is a systemic chronic autoimmune disease with articular manifestations [1]. The manifestations occur due to the increase of pro inflammatory cytokines and it is characterized by joint damage and deformities [1]. In addition, RA patients show extra-articular manifestations as a decrease of muscle mass and physical function [2, 3]. Cruz-Jentoft et al later corrected the cutoffs for ASM/height2 error in women in their study, but this did not compromise their findings [4]. It is known that the inflammatory process has an important role in changes in body composition and physical function in RA patients [5–7]. Additionally, the limitation in physical function may lead to social impairment [7, 8].

Often, in clinical practice and in research, physical function is evaluated by The Health Assessment Questionnaire scores (HAQ) in RA patients [9]. In RA, physical function is generally 24–34% worse than in healthy controls [10]. Lemmey et al. [10] demonstrated that RA patients also had higher "The multidimensional health assessment questionnaire" (MDHAQ) scores than the elderly control group [10]. Also, RA patients not in remission showed higher disability by MDHAQ scores compared to RA patients in remission. Giles et al [11] showed that HAQ is inversely related to appendicular lean mass, and directly related to appendicular fat mass. Although the HAQ demonstrates associations with disease activity, muscle strength and physical performance in RA patients [12], it does not include mental and social aspects.

In accordance with The World Health Organization (WHO), mental, social and physical well-being lead to improvement of health status [13]. To promote this integration, the WHO developed a tool called the International Classification of Functioning, Disability and Health (ICF) [14]. Thus, the ICF offers a biopsychosocial approach that presents an outcome called functionality [15, 16]. Functionality is defined as a dynamic interaction between a person's health status and environmental and personal factors [16].

This classification may be applied in diverse health conditions such as rheumatoid arthritis and other rheumatic diseases, post-acute stroke and chronic obstructive pulmonary disease [17, 18]. In patients with RA, the group of specific ICF codes was created and validated, called the ICF Core Set [19]. Stuck et al [20] proposed a total of 96 categories with 25 subcategories of the component body functions, 18 from body structures, 32 from activities and participation and 21 of environmental factors.

Thus, the CIF core set for AR is still little explored. No other exploratory studies similar to this one were found in the literature. In this sense, the aim of this study was to evaluate the functionality of a cohort of RA patients using the ICF core set for RA over 12 months.

## Methods

### Study design and patients

We used prospective longitudinal data to perform a prospective cohort study among a well-characterized group of patients with RA. We conducted this study in accordance with STROBE guidelines [21]. The data collection was performed at a tertiary public hospital in Rio Grande do Sul, Brazil (Hospital de Clínicas de Porto Alegre, HCPA) between June 2015 and July 2017. Ninety patients diagnosed with RA according to the ACR/EULAR criteria [22] and aged 40–70 years were enrolled at baseline. The exclusion criteria were dysphagia, illicit drug use, alcohol abuse, severe heart failure (New York Heart Association (NYHA) class III or IV), severe chronic obstructive pulmonary disease, abnormal hepatic function, uncontrolled diabetes (fasting glucose > 140 mg/dL or any random glucose level > 200 mg/dL), thyroid dysfunction (hypo or hyperthyroidism), severe kidney disease (glomerular filtration rate < 15 mL/min), and any other diffuse connective tissue disease. Patients with malignant disease,

deformities in the lower limbs, and any surgical history in the previous year were also excluded. The institutional review board of the Universidade Federal do Rio Grande do Sul, Hospital de Clínicas de Porto Alegre, Brazil (registered under number 15–0297) approved, the declaration of Helsinki principles were followed and all subjects gave written informed consent.

## Variables and measurements

Patients were evaluated at baseline and after 12 months of follow-up. Age, disease duration (years), current smoking, erosions, disease activity, functional test, physical disability and the physical activity were assessed. These data were used to base the RA ICF core set classification.

**Disease activity.**   Each patient's disease activity was measured by the Disease Activity Score-28 (DAS28-CRP) [23] and categorized as remission ($\leq$ 2.6), low (2.6 $>$ and $\leq$ 3.2), moderate (3.2 $>$ and $\leq$ 5.1) or high ($>$ 5.1) [23, 24].

**Physical disability.**   The physical disability was assessed by the Health Assessment Questionnaire-Disability Index (HAQ-DI) [25]. The HAQ-DI comprises eight categories and the sum of scores is then divided by the number of categories, yielding a total score ranging from 0 (best) to 3 (worst). The RA patients was categorized in mild disability (HAQ scores 0–1), moderate disability (HAQ scores 1–2) and severe disability (HAQ scores 2–3).

**Physical activity.**   The Physical activity was assessed by the Physical Activity Questionnaire (IPAQ) [26]. Patients were classified as: 1—"low level of physical activity", when no physical activity was reported or did not fit into the other categories; 2—"moderate level of physical activity", when 3 or more days of vigorous activity of at least 20 minutes a day or 5 or more days of moderate-intensity activity or walking for at least 30 minutes a day or 5 or more days of any combination of walking, moderate intensity or vigorous intensity activities reaching a minimum of at least 600 MET-min / week; 3- "high level of physical activity", when activity of vigorous intensity for at least 3 days and accumulating at least 1,500 MET- minutes / week or 7 or more days of any combination of walking, moderate intensity or vigorous intensity activities reaching a minimum of at least 3,000 MET-minutes / week.

**Handgrip strength.**   Handgrip strength was measured using a handheld dynamometer (Jamar Hydraulic Hand Dynamometer, Preston, USA). The patient was instructed to squeeze the handle as hard as possible for 5 s, and the maximal isometric voluntary contraction (MIVC). The cutoff point used to define muscle weakness was established by Cruz-Jentoft et al [4], being $<$27kg for men and $<$16kg for women.

**Fatigue and anorexia.**   Fatigue was measured using the Functional Assessment of Chronic Illness Therapy Fatigue (FACIT-F) scale, validated for Portuguese. The sum score ranges from 0 to 52, with lower scores indicating greater fatigue. Scores $\leq$ 20 characterize fatigue [27]. Anorexia was assessed using the anorexia/cachexia section of the Functional Assessment of Anorexia/Cachexia Therapy (FACCT) questionnaire, validated for Portuguese. The sum score ranges from 0 to 48, with lower scores indicating less appetite. Anorexia was defined as a FAACT score $\leq$24 [28].

**Functional test.**   The test consists of performing the movement of getting up from a chair until standing in an upright posture and returning to a sitting position [29, 30]. The two forms of this assessment were performed, in the sit and stand test of 5 repetitions the time it takes the patient to perform 5 repetitions was evaluated and in the sit and stand test of 30 seconds the number of repetitions of this action in 30 seconds was evaluated. The functional test used was the 5-repeat stand-and-sit test and the 30-second sit-and-stand test were collected. In the 5-repeat stand-and-sit test, fast time $<$12 seconds, intermediate time 12 to 15 seconds, slow$>$ 15 seconds and unable to perform the test were considered as unable [30]. In the 30-second sit

**Table 1. Qualifier and level of commitment.**

| Code with qualifier | Descriptive Commitment Level | Quantitative Commitment Level |
|---|---|---|
| .0 | There is NO limitation | 0–4% |
| .1 | MILD limitation | 5–24% |
| .2 | MODERATE limitation | 25–49% |
| .3 | SEVERE limitation | 50–95% |
| .4 | COMPLETE limitation | 96–100% |
| .8 | not specified | - |
| .9 | not applicable | - |

and stand test, there is no consensus on the cutoff point for the population with RA or the elderly.

**ICF classification.** The International Classification of Functioning and Health was used with collection of secondary data sources (database) as the practical manual for the use of the International Classification of Functioning, Disability and Health (ICF) written by the World Health Organization (WHO) allows [16].

ICF codes require the use of qualifiers that denote the magnitude or severity of the problem in question and are represented by numbers from 0 to 4. For example, if we are evaluating component B130 which represents the "energy and activation functions" of the BODY FUNCTIONS domain and we want to represent that this function is affected from 96% to 100% or we mean that the patient has complete impairment of function, we represent as follows: B130.4. To quantify the level of functional impairment in ICF codes, we use the criteria in Table 1.

**ICF domains.** Within the domains of the ICF, we choose to classify the cohort of patients with RA into "Body Functions", domain represented by the letter "B" and "Activities and Participation" represented by the letter "D". We excluded other domains and subdomains so that it was possible to perform the collection in secondary data sources (database).

## Statistical methods

Paired Student's t-test was used to compare demographic variables and clinical characteristics between baseline and 12-month assessments. To assess the change in ICF codes from baseline and within 12 months, we used the McNemar test. We used the delta of the means of the ICF codes in the Spearman test to correlate the change in the ICF codes with the deltas of the means of DAS28, IPAQ and HAQ. To associate the ICF codes with smoke and erosions we used the chi-square test and to associate it with the time of diagnosis we used the Mann Whitney test. The level of significance was set at $p \leq 0.05$ for all analyzes and the strength of the correlation was considered based on the study by Walpole, R. E et al [31]. Statistical analyzes were performed using PASW 18.0 Statistics for Windows.

## Results

### Demographic and clinical characteristics

Of the 90 RA patients recruited, 81 completed the 1-year follow-up. Of the nine dropouts, two patients died (an unreported reason for death and a stroke), five withdrew consent, and two moved out of the study region or could not be contacted (Fig 1).

At baseline, the mean (SD) age of patients was 56.5 years (7.3), while the median (IQR) disease duration was 8 years (3–18). Most patients were women (78/90; 86.7%). The median

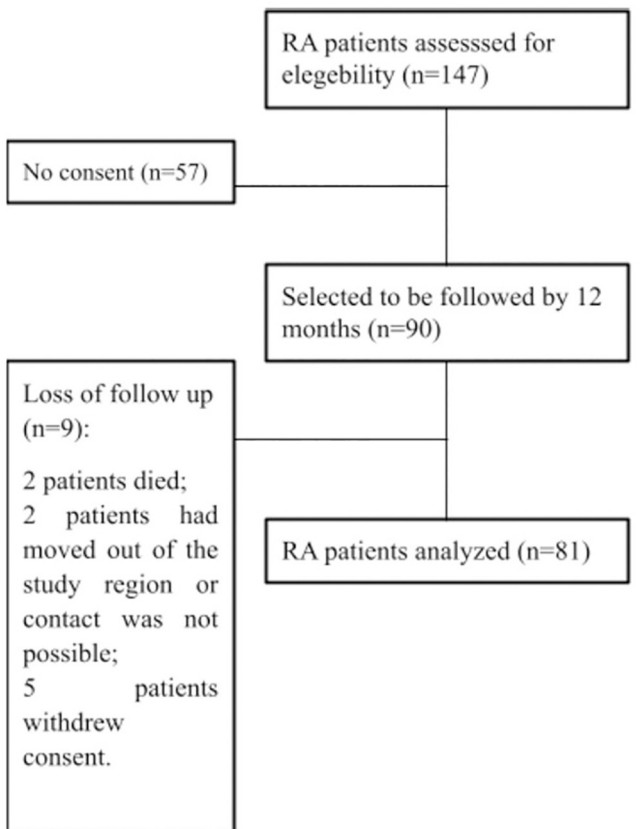

**Fig 1. Flow diagram of the study population.**

DAS28-CRP score was 3.0 (1.0–3.0). Twenty-four patients (27%) were treated with bDMARDs at baseline, 73.3% (84/90) were on conventional synthetic DMARDs (csDMARDs) and 58.9% (45/90) were on glucocorticoids. During follow-up, one patient discontinued the bDMARD and four patients started the use of bDMARDS. After 12 months, we didn't note any statistically significant difference of change of mean DAS28-CRP (p > 0.05) among patients that continued bDMARDS therapy, patients that started bDMARDS therapy during follow-up and patients that continued csDMARDS therapy. There was no statistically significant difference in the demographic and clinical characteristics baseline and 12-month data Table 2.

## The ICF codes

In total, there were 15 components to be evaluated in the "BODY FUNCTIONS" domain represented by the letter "b", only 8 were evaluated because the others did not have data in this cohort that represented the component. For example, the component "b430 Haematological system functions" did not have data to represent it in this cohort and so was left out of the analysis. The same happened with the domain "ACTIVITIES AND PARTICIPATION", represented by the letter "d", which has a total of 32 components and only 6 components were evaluated. Then, from the total of 47 codes in each domain, only 14 were analyzed.

At baseline, the highest frequency of patients was in the "mild limitation" quantifier (n = 326; 28.7%) and the lowest frequency was in the "complete limitation" quantifier (n = 82; 7.2%). The frequency of code distribution in the baseline is detailed in Table 3.

**Table 2. Demographic and clinical characteristics of the study sample.**

| | Baseline (n = 90) | At 12 months (n = 81) | p value |
|---|---|---|---|
| **Age (years), mean ± SD** | 56.5 ± 7.3 | | |
| **Disease duration (years), median (IQR)** | 8.5 (3.0–18.0) | | |
| **Women, n (%)** | 78 (86.7) | 72 (88.9) | |
| **Men, n (%)** | 12 (13.3) | 9 (11.1) | |
| **Current smoker, n (%)** | 18 (20.0) | | |
| **Disease activity** | | | |
| DAS28-CRP, median (IQR) | 3.0 (1.0–3.0) | 3.0 (2.0–3.0) | 0.353* |
| Remission, n (%) | 25 (27.8) | 18 (20.0) | 0.161** |
| Low, n (%) | 8 (8.9) | 11 (12.2) | |
| Moderate, n (%) | 31 (34.4) | 34 (37.8) | |
| High, n (%) | 19 (21.1) | 15 (16.7) | |
| CRP, median (IQR) | 4.15 (3.0–9.9) | 3.8 (1.3–10.3) | 0.209* |
| **Positive rheumatoid factor, n (%)** | 77 (85.6) | | |
| **Positive ACPA, n (%)** | 34/45 (75.5) | | |
| **Presence of erosion, n (%)** | 64 (71.1) | | |
| **Physical disability (HAQ-DI)** | 1.2 ± 0.1 | 1.1 ± 0.1 | 0.100* |
| **Treatment regimen** | | | |
| **MTX monotherapy, n (%)** | 52 (57.8) | 39 (48.1) | |
| **MTX with concurrent csDMARD, n (%)** | 14 (100.0) | 15 (18.5) | |
| **MTX dose (mg/week), median (IQR)** | 20.0 (15.0–25.0) | 20.0 (15.0–25.0) | 0.355[a] |
| **bDMARDs, n (%)** | 27 (30.0) | 29 (32.2) | |
| **Glucocorticoids, n (%)** | 53 (58.9) | 41 (50.6) | |
| **Glucocorticoid dose (mg/day), median (IQR)** | 5.0 (5.0–10.0) | 5.0 (2.5–10.0) | |
| **Physical activity (IPAQ)** | | | |
| Low, n (%) | 30 (33.3) | 19 (23.5) | 0.121[b] |
| Moderate, n (%) | 43 (47.8) | 44 (54.2) | |
| High, n (%) | 17 (18.9) | 18 (22.3) | |

DAS28-CRP, the Disease Activity Score-28 with C reactive protein; CRP, C reactive protein; ACPA, anti-citrullinated protein antibodies; DAS28-CRP, the Disease Activity Score-28 with C reactive protein; CRP, C reactive protein; MTX, methotrexate; csDMARD, conventional synthetic disease-modifying antirheumatic drugs (methotrexate, leflunomide, hydroxychloroquine, and sulfasalazine); bDMARDs, biologic disease-modifying antirheumatic drugs (adalimumab, etanercept, infliximab, certolizumab, golimumab, rituximab, tocilizumab, and abatacept); HAQ-DI, the Health Assessment Questionnaire-Disability Index; IPAQ, International Physical Activity Questionnaire.

[a]Pairwise Student's t test

[b]Pearson Chi-square

The code B455 showed a statistically significant improvement (Fig 2; p <0.02) after 12 months, while the code B730 showed a statistically significant decrease (Fig 2; p<0.00) after 12 months. No other code has demonstrated significant change from baseline to after 12 months.

Correlations among ICF and disease activity physical activity levels, physical function, smoking, disease duration and erosions.

The decrease in the activity and participation domain (D) is associated with worsening in the Disease Activity Score-28 (DAS28-CRP) (r = 0.271;p = 0.022). The others ICF codes did not show a significant correlation with the Physical Activity Questionnaire (IPAQ) or the Health Assessment Questionnaire-Disability Index (HAQ-DI). In addition, no statistically significant association was found between the worsening of the ICF codes (qualifier increase

**Table 3. The frequency of code distribution in the baseline.**

| CODES | NO limitation (n;%) | MILDlimitation (n;%) | MODERATE limitation (n;%) | SEVERE limitation (n;%) | COMPLETE limitation (n;%) |
|---|---|---|---|---|---|
| B130 | 21(25.9) | 28(4.6) | 14 (17.3) | 14 (17.3) | 4 (4.9) |
| B134 | 9 (11.1) | 12 (14.8) | 17 (21) | 30 (37) | 13 (16) |
| B280 | 15 (18.5) | 16 (19.7) | 26 (32) | 20 (24.7) | 4 (4.9) |
| B455 | 6 (8) | 23 (29.1) | 22 (27.8) | 23 (29.1) | 5 (6.3) |
| B510 | 0 (0) | 35 (43.2) | 41 (50.6) | 5 (6.7) | 0 (0) |
| B640 | 24 (29.6) | 9 (11.1) | 11 (13.5) | 25 (30.9) | 12 (14.8) |
| B730 | 40 (50) | 10 (12.5) | 12 (15) | 13 (16.2) | 5 (6.2) |
| B740 | 32 (39.5) | 30 (37) | 6 (7.4) | 4 (4.9) | 9 (11.1) |
| D230 | 26 (32) | 16 (19.7) | 14 (17.3) | 14 (17.3) | 11 (13.6) |
| D550 | 60 (74) | 7 (8.6) | 8 (9.9) | 4 (4.9) | 2 (2.5) |
| D760 | 11 (13.6) | 25 (30.9) | 30 (37) | 15 (18.5) | 0 (0) |
| D770 | 17 (21) | 47 (58) | 5 (6.2) | 6 (7.4) | 6 (7.4) |
| D850 | 9 (11.1) | 24 (29.6) | 17 (21) | 24 (29.6) | 7 (8.6) |
| D920 | 14 (17.2) | 44 (54.3) | 9 (11.1) | 10 (12.3) | 4 (4.9) |
| TOTAL | 267 (23.5) | 326 (28.7) | 232 (20.4) | 187 (16.4) | 82 (7.2) |

B130, Energy and drive functions; B134, Sleep functions; B280, Sensation of Pain; B455, Exercise tolerance functions; B510, Ingestion functions; B640, Sexual functions; B730, Muscle power functions; B740, Muscle endurance functions; D230, Carrying out daily routine; D550, Eating; D760, Family relationships; D770, Intimate relationships; D850, Remunerative employment.

from 0–4) and smoking (p = 0,247), time from diagnosis (p = 0,802) and erosions (p = 0,479). These results are summarized in Table 4.

## Discussion

Our study demonstrated that more patients had mild to moderate limitation quantifiers of the ICF Core set for RA at baseline. In addition, to exercise tolerance(B455) code the more patients

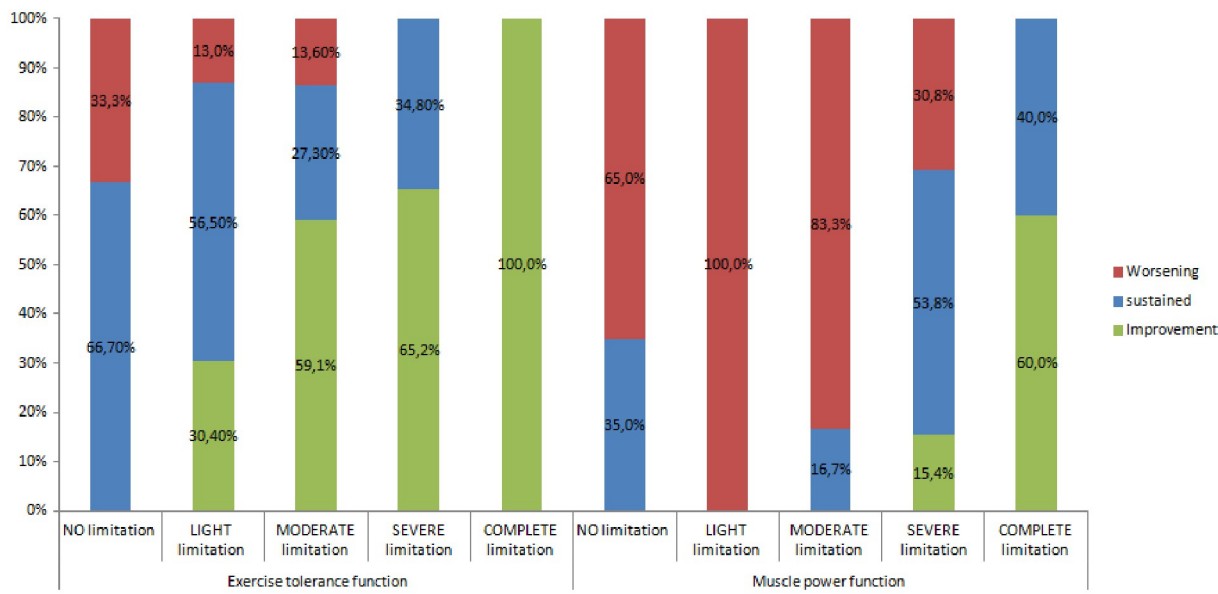

**Fig 2. Change in exercise tolerance functions and power muscle functions over 12 months.**

**Table 4. Correlations of ICF codes, DAS 28, IPAQ and HAQ-DI.**

|  | N | Correlation Coefficient | P |
|---|---|---|---|
| ΔICFb x ΔDAS28 | 72 | 0.098 | 0.411 |
| ΔICFd x ΔDAS28 | 72 | 0.271[a] | 0.022[b] |
| ΔICFb x ΔIPAQ | 80 | 0.126 | 0.267 |
| ΔICFd x ΔIPAQ | 80 | -0.190 | 0.092 |
| ΔICFb x ΔHAQ-DI | 80 | 0.101 | 0.374 |
| ΔICFd x ΔHAQ-DI | 80 | 0.818 | 0.108 |

Δ = final mean—means initial; ICFb, Domain Functions of the Body of the International Classification of Functioning and Health; ICFd, Domain activity and participation of the International Classification of Functioning and Health DAS28, Disease Activity Score-28; IPAQ, Physical Activity Questionnaire; HAQ-DI, Health Assessment Questionnaire-Disability Index

[a]Correlation is significant at the 0.05 level, Spearman's test.

[b]Statistically significant.

had mild to severe limitations and significantly improved over 12 months. On the other hand, muscle strength (B730) code showed worsening over 12 months. Another important point of the study was the negative relationship between the disease activity status and the activity and participation domain. As far as we know, this was the first study to classify ICF codes based on secondary codes (database with clinical data and physical assessments) in a prospective cohort of patients with rheumatoid arthritis.

It is well known that RA patients have reduced functional capacity compared to controls. Functional capacity is defined as what an individual can do in a 'standardized' environment, usually in a test and adjusted situation [32–34]. In addition, functional capacity can negatively affect physical performance and quality of life [32, 33]. Physical performance refers to uncontrolled situations and environments, without human assistance and that do not involve test situations, would be daily activities [34]. However, in our study, the RA patients showed mild to moderate functional capacity limitations. In addition, Rosa-Gonçalves et al [33] pointed out the worsening of functional capacity and quality of life at higher levels of disease activity. Corroborating with Rosa-Gonçalves et al [33], we found that disease activity status is related to decreased activity and participation of RA patients.

Our findings report an increase in exercise tolerance and a decrease in muscle strength in patients with RA. Therefore, we believe that by starting the study with low disease activity, this factor positively influences functional limitation. Due to this, the improvement in exercise tolerance can be explained. Another important point is that we used the handgrip test to measure the muscle strength of patients and EWGSOP recommends using this test for a reliable strength measurement for RA patients [4]. Although the patients had low disease activity, muscle strength was reduced. This reduction can be explained by the aging process. Physiological changes such as loss of motor units, changes in fiber type, muscle fiber atrophy, and reduced neuromuscular activation can affect the speed and strength of movements [35]. Architectural changes with aging include, among others, an increase in fat infiltration into skeletal muscle and a change in the elastic fiber system [36]. Another change that occurs due to age is the decline in muscular aerobic capacity [37]. Furthermore, aging has been shown to require significantly greater activation of various motor areas of the brain to perform the same motor prehension task as younger adults [35].

To prove that the ICF Core Set is indeed responsive, Uhlig et al [38] evaluated the responsiveness of the ICF Core Set for RA and found that there is moderate responsiveness.

Therefore, the ICF Core Set for AR allows observing and reporting small changes in the patient. This finding confirms that our results are reliable, because, even if the changes were subtle throughout the study, the ICF Core Set for RA would be sensitive enough to assess them.

Our main limitation was the inability to assess the 47 codes of the two domains of activity and participation and bodily functions, as our study was not designed from the ICF. This shows us that the functionality assessment by the ICF takes into account aspects that we often do not collect in clinical research, even with broad assessments and using gold standard measurements. Codes important to function are generally not taken into account in studies involving physical assessments, such as b1801 Body image, b715 Stability of joint functions, d470 Use of transport, among others.

We conclude that relevant aspects of the ICF Core Set for RA were able to adequately express the physical and functional factors of the RA cohort. This tool allows a common language for the interdisciplinary team, which can enhance the use of timely interventions to prevent physical disability in clinical practice. Thus, it is potentially useful in assessing the functionality of patients with RA, but further studies are needed to be able to use all domains of the ICF Core Set for AR.

## Acknowledgments

Thanks to all colleagues at LABDAI for their continued help, discussions, criticism and attention and that made the work environment very productive. in particular to the clinical research team with whom I could and also the HCPA statistical team with whom I had contact several times! We thank the Fundo de Incentivo à Pesquisa e Eventos of Hospital de Clínicas de Porto Alegre for their support. To the Federal University of Rio Grande do Sul. To the Rheumatology Service of Hospital de Clínicas de Porto Alegre.

## Author Contributions

**Conceptualization:** Juliana Portes.

**Data curation:** Rafaela C. E. Santo.

**Formal analysis:** Juliana Portes.

**Funding acquisition:** Claiton V. Brenol.

**Investigation:** Juliana Portes.

**Methodology:** Juliana Portes.

**Project administration:** Ricardo M. Xavier.

**Supervision:** Rafaela C. E. Santo, Claiton V. Brenol.

**Visualization:** Rafaela C. E. Santo, Ricardo M. Xavier.

**Writing – original draft:** Juliana Portes.

**Writing – review & editing:** Juliana Portes.

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
