## [Decision Letter · Decision Letter 0]

10 Jan 2023

PONE-D-22-32670Funcionality Assessed by the Core Set of the International Classification of Functionality and Health for Rheumatoid Arthritis: A cohort study.PLOS ONE

Dear Dr. Portes,

Thank you for submitting your manuscript to PLOS ONE. After careful consideration, we feel that it has merit but does not fully meet PLOS ONE’s publication criteria as it currently stands. Therefore, we invite you to submit a revised version of the manuscript that addresses the points raised during the review process.

We look forward to receiving your revised manuscript.

Kind regards,

Soham Bandyopadhyay

Academic Editor

PLOS ONE

Journal Requirements:

"https://www.gov.br/capes/pt-br

https://www.fipe.org.br/pt-br/ensino/extensao-curta-duracao/

This study was funded by the Research Incentive Foundation (FIPE) of the Hospital de Clínicas de Porto Alegre and the Coordination for the Improvement of Higher Education Personnel (CAPES).

FIPE was used to cover general research expenses and a CAPES grant was given to study author Juliana Portes. Funders did not participate in any stage of the study."

3. Thank you for stating the following in the cover letter Section of your manuscript: 

"This study was funded by Research Incentive Foundation (FIPE) of the Hospital de Clínicas de Porto Alegre and Coordination for the Improvement of Higher Education Persons (CAPES)."

"https://www.gov.br/capes/pt-br

https://www.fipe.org.br/pt-br/ensino/extensao-curta-duracao/

This study was funded by the Research Incentive Foundation (FIPE) of the Hospital de Clínicas de Porto Alegre and the Coordination for the Improvement of Higher Education Personnel (CAPES).

FIPE was used to cover general research expenses and a CAPES grant was given to study author Juliana Portes. Funders did not participate in any stage of the study."

**Comments to the Author**

1. Is the manuscript technically sound, and do the data support the conclusions?

Reviewer #1: Partly

2. Has the statistical analysis been performed appropriately and rigorously? 

Reviewer #1: Yes

3. Have the authors made all data underlying the findings in their manuscript fully available?

Reviewer #1: Yes

4. Is the manuscript presented in an intelligible fashion and written in standard English?

Reviewer #1: Yes

5. Review Comments to the Author

Reviewer #1: This is an interesting informative manuscript on RA functionality assessment.

1. Typographical errors throughout the manuscript should be corrected, for example, in the Abstract- under "Objective" .....to evaluate "function" and not the functioning; under the" Results" -......over 12 months. The and not" the" . On page 4, " sence' should read; ' sense'

2. The conclusions in the Abstract and at the end of the article should be rewritten to be illustrative of the results.

3. The same font should be used throughout the manuscript.

4. The word: " light" throughout the manuscript and in the Tables should be replaced by: " mild".

5. The Discussion section is somewhat weak-authors need to elaborate on their findings; they need to explain their results more comprehensively and defend their findings.

6. PLOS authors have the option to publish the peer review history of their article (what does this mean?). If published, this will include your full peer review and any attached files.

Reviewer #1: No

---

## [Author Response · Author response to Decision Letter 0]

23 Apr 2023

The formatting of titles, subtitles and tables were changed to follow the journal's rules. All mentions of funders were removed from the article, however we had to include the phrase "We thank the Fundo de Incentivo à Pesquisa e Eventos of Hospital de Clínicas de Porto Alegre for their support." in the acknowledgments so that the value of the publication is subsidized and the role of the funders was stated in the cover letter. We have modified our conclusion and amended our discussion to follow the reviewer's guidance.

---

## [Editor Report · Decision Letter 1]

15 May 2023

Funcionality Assessed by the Core Set of the International Classification of Functionality and Health for Rheumatoid Arthritis: A cohort study.

PONE-D-22-32670R1

Dear Dr. Portes

We’re pleased to inform you that your manuscript has been judged scientifically suitable for publication and will be formally accepted for publication once it meets all outstanding technical requirements.

Kind regards,

Soham Bandyopadhyay

Academic Editor

PLOS ONE
---

## [Editor Report · Acceptance letter]

19 May 2023

PONE-D-22-32670R1 

Funcionality Assessed by the Core Set of the International Classification of Functionality and Health for Rheumatoid Arthritis: A cohort study. 

Dear Dr. Portes:

I'm pleased to inform you that your manuscript has been deemed suitable for publication in PLOS ONE. Congratulations! Your manuscript is now with our production department. 

Kind regards, 

on behalf of

Dr. Soham Bandyopadhyay 

Academic Editor

PLOS ONE